# Age-Related Mitochondrial Alterations Contribute to Myocardial Responses During Sepsis

**DOI:** 10.3390/cells14151221

**Published:** 2025-08-07

**Authors:** Jiayue Du, Qing Yu, Olufisayo E. Anjorin, Meijing Wang

**Affiliations:** 1Center for Surgical Sciences, Department of Surgery, Indiana University School of Medicine, Indianapolis, IN 46202, USA; jiaydu@iu.edu (J.D.); qyu@iu.edu (Q.Y.); 2Division of Cardiothoracic Surgery, Department of Surgery, Indiana University School of Medicine, Indianapolis, IN 46202, USA; panjorin@iu.edu

**Keywords:** mitochondrial metabolic function, septic cardiomyopathy, OXPHOS, aging, inflammation

## Abstract

Sepsis-induced myocardial injury is age-related and leads to increased mortality. Considering the importance of mitochondrial dysfunction in cardiac impairment, we aimed to investigate whether aging exacerbates the cardiac mitochondrial metabolic response to inflammation, thus leading to increased cardiac dysfunction in the elderly. Cecal ligation and puncture (CLP) was conducted in young adult (12–18 weeks) and aged (19–21 months) male C57BL/6 mice. Cardiac function was detected 20 h post-CLP. Additionally, cardiomyocytes isolated from young adult and aged male mice were used for assessments of mitochondrial respiratory function +/– TNFα or LPS. Protein levels of oxidative phosphorylation (OXPHOS), NADPH oxidase (NOX)2, NOX4, phosphor-STAT3 and STAT3 were determined in mouse hearts 24 h post-CLP and in cardiomyocytes following inflammatory stimuli. CLP significantly reduced cardiac contractility in both young and aged mice, with a higher incidence and greater severity of cardiac functional depression in the older group. Mitochondrial respiratory capacity was decreased in cardiomyocytes derived from aged mice, with increased susceptible to inflammatory toxic effects compared to those from young adult mice. The age-dependent changes were observed in myocardial OXPHOS complexes and NOX4. Importantly, CLP led to a significant increase in OXPHOS protein levels in the hearts of older mice, suggesting a possible compensatory response to decreased mitochondrial metabolic function and a greater potential for reactive oxygen species (ROS) generation. Our findings highlight that the response of aging-impaired mitochondria to inflammation may underlie the worsened cardiac functional depression in the aged group during sepsis.

## 1. Introduction

Sepsis is a life-threatening syndrome with extremely high mortality, affecting 49 million people annually and accounting for 11 million deaths—about 20% of global mortality [1]. Sepsis-induced cardiac damage or dysfunction, considered septic cardiomyopathy, is a common and severe complication of post-sepsis multi-organ dysfunction [2,3], significantly impacting patient prognosis [4,5,6]. Accumulated evidence has suggested that cardiac dysfunction during sepsis is associated with high mortality and poor prognosis. In addition, at least one in two sepsis patients suffers from impaired heart function [7]. Of note, mitochondria are essential intracellular organelles that maintain cardiovascular cell homeostasis and serve as the primary site of ATP production via oxidative phosphorylation (OXPHOS) [8,9]. In addition, they play a key role in regulating intracellular reactive oxygen species (ROS) levels in cardiomyocytes, particularly during injury [10]. Mitochondrial dysfunction and oxidative stress are major contributors to the loss of cardiac function after cardiac injury. The critical role of mitochondrial impairment has been well-documented in many sepsis studies. Importantly, mitochondrial disruption and metabolic shutdown have been identified as key drivers in the development of sepsis-induced organ failure. Calcium overload altered mitochondrial enzyme activity, decreased ATP production, and mtDNA production following ROS attack on mitochondria in sepsis can worsen disturbances in mitochondrial energy metabolism [11,12]. Clinically, mitochondrial dysfunction is related to poor prognosis in sepsis [13,14]. Given that mitochondria account for approximately one-third of the volume of cardiomyocytes, and that maintaining their integrity and function is essential for preventing cardiovascular and metabolic diseases, preserving mitochondrial health may represent a potential therapeutic target for septic cardiomyopathy.

It is worth noting that cardiovascular disease and mitochondrial dysfunction both increase with age [15,16]. Older adults have a 13.1-fold higher risk of developing sepsis and a 1.56-fold higher mortality rate compared to individuals under 65 years of age [17]. Age-related declines in immune responsiveness and increased oxidative stress following sepsis may contribute to more severe infections and higher mortality. Sepsis-induced myocardial injury is also age-dependent and likely plays a role in the observed differences in clinical outcomes across age groups [18]. Both intrinsic and extrinsic factors can compromise mitochondrial function, contributing to the development of aging-associated heart disease [19]. However, it remains unclear whether aging-related mitochondrial dysfunction directly increases susceptibility to cardiac dysfunction during sepsis in the elderly.

Under stress, aged cardiac cells are more likely than young adult cells to produce intracellular ROS [20]. The resulting oxidative damage accumulates over time. Importantly, senescent mitochondria lose their ability to effectively scavenge ROS, which can further damage and ultimately destroy mitochondrial structure [21], causing cardiac damage/dysfunction. Of note, the OXPHOS pathway, a major source of ROS, is significantly affected by the aging process [22,23,24,25]. In the present study, we aimed to determine whether aging affects the metabolic response of cardiac mitochondria to inflammatory mediators and alters OXPHOS signaling during sepsis, using a mouse model of cecum ligation and puncture (CLP), the most widely accepted sepsis model [26,27], to mimic the clinical situation of septic patients. This may provide deeper insight into disease mechanisms in the aging population with sepsis and help identify potential therapeutic targets to reduce morbidity and mortality in elderly patients with severe sepsis.

## 2. Materials and Methods

### 2.1. Animals

Male C57BL/6J mice were obtained from the Jackson Laboratories (Bar Harbor, ME, USA). Studies from our group [28] and others [29] have demonstrated sex-specific differences in myocardial responses to sepsis. To minimize variability, we utilized male mice in the present work. All mice were housed in the Laboratory Animal Research Facility at Indiana University School of Medicine for at least 5 days before experiments, under controlled temperature and humidity conditions with a 12 h light/dark cycle and free access to food and water. Animal work in this study was reviewed and approved by the Institutional Animal Care and Use Committee of Indiana University. All animals received humane care in compliance with the National Research Council’s *Guide for the Care and Use of Laboratory Animals*.

### 2.2. CLP-Induced Sepsis Model

CLP and sham operation were performed in young (12–18 weeks) and aged (19–21 months) male mice. Polymicrobial sepsis was induced using CLP [30]. Mice were anesthetized by inhalation of isoflurane and hair was shaved at the abdomen. A midline incision (<2 cm) was made to expose the cecum. The cecum was ligated with a 4.0 silk suture at one-third of its total length from the tip and perforated with a 20-gauge needle between the ligation site and the tip to extrude a small amount of feces. The cecum was placed back into the abdomen and the abdomen was closed double. Sham-operated mice were subjected to the same surgical procedure without CLP. A single dose of sterile saline was administrated peritoneal immediately post-operation. Mice were returned to their cages with food and water ad libitum. Heart tissue was collected at 24 h post-operation for subsequent assessments.

### 2.3. Echocardiographic Evaluation

The Vevo^®^ 2100 system (Fujifilm VisualSonics Inc., Toronto, ON, Canada) was used to evaluate cardiac function on animals 20 h post-surgery. Experimental animals were anesthetized with isoflurane (induction 5%, maintenance 1–2%) and placed in a supine position on the heated plate (the temperature was maintained at 36.5 °C ± 0.5 °C). Cardiac function was assessed with a heart rate maintained between 400 and 500 beats per minute. M-mode scanning of the left ventricular chamber was used for analysis of left ventricular (LV) ejection fraction (EF), fractional shortening (FS). The echocardiograms were performed in a blinded fashion.

### 2.4. Western Blotting

Heart tissue or cardiomyocytes were lysed with RIPA buffer containing Halt™ Protease and Phosphatase Single-Use Inhibitors Cocktail (Thermo Fisher Scientific, Waltham, MA, USA), and the supernatant was collected. Protein concentration was determined by the BCA method, and equal amounts of protein were electrophoresed in 4–20% Criterion™ TGX™ Precast Midi Protein Gel (Bio-Rad, Hercules, CA, USA) and transferred to nitrocellulose membranes. The membranes were blocked with 5% non-fat milk in TBST at room temperature for 1 h, and then incubated overnight at 4 °C with primary antibodies as follows: OXPHOS (#45-8099, Thermo Fisher Scientific), GAPDH (#2118, Cell Signaling, Danvers, MA, USA), Stat3 (#12640, Cell Signaling), Phospho-Stat3 (Tyr705) (#9131, Cell Signaling), NOX2 (19013-1-AP, Proteintech, Rosemont, IL, USA), NOX4 (PA5-72816, ThermoFisher Scientific). After that, membranes were incubated with HRP-conjugated secondary antibody for 1 h at room temperature. A ChemiDoc system (Bio-Rad) was used to detect immunoblotting bands, which were quantified using the Image J software (Version 1.51, NIH).

### 2.5. Transmission Electron Microscopy (TEM)

TEM was performed to examine the ultrastructure of the heart tissue. As we described previously [31], cardiac tissues were first fixed with 3% glutaraldehyde in 0.1 M sodium cacodylate (SC) buffer and then incubated with 1% osmium tetroxide in 0.1 M SC buffer for 1 h. After that, the tissue was dehydrated, infiltrated, and embedded. Thin sections were cut, stained, and viewed on a Tecnai Spirit (ThermoFisher, Hillsboro, OR, USA). Digital images were blindly taken using an AMT (Advanced Microscope Techniques, Danvers, MA, USA) CCD camera. Mitochondrial number and size in each field of view were analyzed using Image J software (NIH).

### 2.6. Mouse Cardiomyocyte Isolation

The Langendorff perfusion system was used to isolate individual cardiomyocytes from young adult (11–21 weeks) and aged male (20–21 months) mouse hearts [31,32]. Briefly, mice were injected with heparin (100 IU, i.p.) and then euthanized with an overdose of isoflurane. The hearts were rapidly removed and placed into a simple Langendorff apparatus via aorta cannulation. The hearts were perfused with a calcium-free perfusion buffer first and then digested with collagenase II (1.5 mg/mL) in perfusion buffer containing 50 mM calcium. Cells were sequentially recovered in calcium-containing buffer (100, 250, 500, or 1000 µmol/L CaCl_2_) and seeded into laminin-precoated 6-well plates. After 2 h cultivation for adherence, cardiomyocytes in the 6-well plates were treated with vehicle, 5 ng/mL TNFα or 5 μg/mL LPS for 1 h and then collected for protein isolation. The cardiomyocytes were also inoculated into laminin (20 mg/mL) pre-coated 96-well plates with cardiomyocyte plating medium (Opti-MEM with glutamine +2.5% FBS, 10 mM BDM, and 1% Pen/Strep) and used for experiments.

### 2.7. Measurement of Mitochondrial Bioenergetic Profiles

The bioenergetic profiles of cardiomyocytes from young and aged mice were assessed using a Seahorse XF96 analyzer (Agilent, Santa Clara, CA, USA), as we previously reported [33]. The mitochondrial oxygen consumption rate (OCR) was calculated in pmol/min. First, the Seahorse Cell Energy Phenotype test was used to simultaneously assess mitochondrial respiration and glycolysis. The cells were incubated with supplemented XF medium (5 mM glucose, 1 mM pyruvate, and 2 mM glutamine) in 37 °C, non-CO_2_ incubator for 1 h, and then measured metabolic phenotypes under basal condition, followed by a simultaneous addition of oligomycin (1 μM) to inhibit ATP synthase and the uncoupling agent Carbonyl cyanide-4 (trifluoromethoxy) phenylhydrazone (FCCP) (1 μM) for stressed condition. Second, cardiomyocytes from young adult and aged mice were treated without or with TNFα (5 and 10 ng/mL) or LPS (2.5, 5, and 10 μg/mL) in supplemented XF medium for 1 h and then conducted for the Seahorse XF Cell Mito stress test. This test consisted of measuring the baseline OCR, ATP-linked respiration by injection of oligomycin (1 μM), maximal OCR by the addition of FCCP (1 μM), and non-mitochondrial respiration by adding electron transfer inhibitors—rotenone and antimycin A (both 1 μM) to completely inhibit electron transfer. The difference between maximal OCR and basal respiration is the respiratory potential (spare respiratory capacity).

### 2.8. Detection of Mitochondrial Membrane Potential

After 2 h cultivation in cardiomyocyte plating medium, isolated cardiomyocytes from aged mice were treated with vehicle, TNFα (5 ng/mL), or LPS (5 μg/mL). Thirty minutes later, a fluorescent probe JC-1 (5 μM, G-Biosciences, St. Louis, MO, USA) was added to cell wells and the cells were incubated for another 30 min. After that, the cells were washed once with the plating medium. Images were taken on these live cells using an Axio Observer Z1 motorized microscope (Zeiss, Oberchoken, Germany) with a 10× objective lens. Mitochondrial membrane potential was analyzed by quantifying the ratio of red and green fluorescence intensity in individual cardiomyocytes using ImageJ (Version 1.51, NIH).

### 2.9. Statistical Analysis

Data were presented as mean ± SEM. Comparisons were performed between or among groups using unpaired *t*-test or two-way analysis of variance (ANOVA) followed by multiple comparison tests. The difference was considered statistically significant when *p* < 0.05. The GraphPad Prism (Version 10.5.0, Graph-Pad, La Jolla, CA, USA) was used for all statistical analyses.

## 3. Results

### 3.1. Worsened Cardiac Function and Altered Mitochondrial OXPHOS in Aged Mice Compared to Young Animals Following CLP

Previous studies have shown that aged mice exhibited higher mortality rates compared to young adults during sepsis [34,35]. In our pilot survival experiments, we found most aged animals died before 30 h post CLP. To facilitate follow-up analyses in both groups, we therefore focused our investigation on cardiac function at 20 h post CLP and on molecular biological assessments of the heart at 24 h post CLP in this study. Notably, in severe sepsis and septic shock, myocardial dysfunction typically manifests as a reduced ejection fraction [36]. In line with it, we found significantly decreased LVEF and LVFS in both young and aged animals with CLP-induced sepsis compared to their sham groups (Figure 1A,B). Although comparable levels of LVEF and LVFS were noticed between the young and aged sham groups, CLP significantly impaired both parameters more severely in aged mice than in young mice (Figure 1B). Specifically, only 3 out of 9 adult mice showed a > 15% depression in LVEF, whereas all aged mice (6 out of 6) exhibited markedly reduced cardiac function following CLP (Figure 1B). These findings suggest that aged mice were more vulnerable to sepsis-induced cardiac dysfunction.

Given that mitochondrial dysfunction is induced by constitutive changes in mitochondrial OXPHOS and that myocardial aging is associated with reduced functional activity of the OXPHOS [37], we investigated the effects of age and CLP on OXPHOS complex proteins. Protein levels of complex IV-MTCO1 and complex I-NDUFB8 were significantly decreased in the hearts of aged mice compared to that in young adult mice without CLP (Figure 1C,D). In young male mice, CLP decreased cardiac complex V-ATP5A, complex III-UQCRC2, complex IV-MTCO1, complex II-SDHB, and complex I-NDUFB8 proteins (Figure 1C,D). Interestingly, in aged mice, the trend was reversed (Figure 1C,D). The levels of these five complex proteins were significantly higher in the hearts of aged mice compared to those of young mice following CLP, suggesting the potentially greater production of ROS in aged mouse hearts in response to CLP-induced sepsis.

### 3.2. Effects of CLP on the Myocardial Mitochondrial Ultrastructure and Biogenesis in Aged Mice

A hallmark of aging may be the accumulation of dysfunctional or impaired mitochondria in cardiomyocytes. Therefore, we next investigated whether CLP changed mitochondrial morphology and biogenesis (the number and mass of mitochondria) in aged mouse hearts. No significant changes in mitochondrial ultrastructure were observed in aged mice after CLP, as TEM revealed intact structures of both interfibrillar mitochondria (IFM) and subsarcolemmal mitochondria (SSM) in the CLP and Sham groups (Figure 2A). In addition, the number of myocardial mitochondria was not significantly different between the Sham and CLP groups for either IFM or SSM (Figure 2B). Furthermore, CLP did not cause mitochondrial swelling or atrophy in the cardiac SSM, as shown by comparable mitochondrial size between the CLP and sham groups (Figure 2C). However, CLP significantly reduced mitochondrial size for IFM in aged hearts (Figure 2C). Our experiments indicated that CLP did not significantly alter mitochondrial ultrastructure but led to a reduction in IFM size in aged mouse myocardium.

### 3.3. Influence of Aging on Cardiomyocyte Energy Phenotype and Mitochondrial Metabolic Function

Mitochondrial bioenergetics was determined in cardiomyocytes from young adult and aged male mice using a Seahorse extracellular flux (XF) technology to measure the mitochondrial oxygen consumption rate (OCR) and extracellular acidification rate (ECAR). We first assessed cellular energetic phenotype in cardiomyocytes isolated from young adult and aged male mice under basal and stressed conditions. We found that both young and aged mouse cardiomyocytes mainly used mitochondria for aerobic respiration (Figure 3A) and aging did not change energetic phenotype of cardiomyocytes. Next, we examined mitochondrial metabolic function in cardiomyocytes. Notably, the addition of the ATP synthase inhibitor oligomycin did not affect OCR (Figure 3B), possibly due to the quiescent state (and therefore low energy demand) of cultured primary cardiomyocytes. Interestingly, after FCCP stimulation, the maximal OCR and spare capacity of mitochondria in aged mouse cardiomyocytes were significantly lower than those in young mouse cardiomyocytes (Figure 3B). Moreover, significantly decreased non-mitochondrial respiration (mitochondrial respiration shut down due to the addition of rotenone and antimycin A) was observed in aged mice than that in young mice (Figure 3B). Notably, the ECAR was comparable between young adult and aged cardiomyocytes under basal conditions (Figure 3C). However, we found significantly reduced ECAR for maximal acidification in aged cardiomyocytes compared to young adult cells (Figure 3C). These results showed that aging impaired mitochondrial respiratory capacity and glycolytic activity in cardiomyocytes.

### 3.4. Aging Decreases Mitochondrial Respiration Function in Cardiomyocytes Exposed to LPS

LPS, the widely used toxin in endotoxemia models, was employed to mimic the acute inflammatory response associated with sepsis [38]. We first stimulated cardiomyocytes from young and aged mice with three different concentrations of LPS (2.5, 5, and 10 μg/mL) for 1 h. The results showed that in cardiomyocytes from young mice, three different concentrations of LPS had no significant effect on the mitochondrial respiratory capacity (Figure 4A). However, in cardiomyocytes from aged mice, 5 and 10 μg/mL LPS significantly reduced the basal mitochondrial OCR and maximal OCR (Figure 4A). Additionally, 5 μg/mL LPS significantly decreased mitochondrial basal, maximal and non-mitochondrial respiratory capacities in cardiomyocytes of aged mice compared to those from young adult mice (Figure 4B). We further observed a similar patten for maximal acidification with a significant decrease in ECAR in aged cardiomyocytes following exposure to 5 μg/mL LPS (Figure 4C). These results suggest that cardiomyocytes from aged mice are more vulnerable to LPS-induced mitochondrial damage and inflammation has a greater negative impact on myocardial metabolism in the elderly.

### 3.5. Aging Worsens TNFα-Damaged Mitochondrial Respiratory Function in Cardiomyocytes

Infectious stimuli induce monocytes/macrophages and other cells to release local and systemic inflammatory mediators, especially tumor necrosis factor α (TNFα) [39]. In septic animals, plasma TNFα levels can exceed 10 ng/mL [40]. In this study, we used TNFα at concentrations of 5 and 10 ng/mL to stress cardiomyocytes, simulating the pro-inflammatory environment observed during sepsis. We found that in young adult mice, TNFα stimulation (5 and 10 ng/mL) had no significant effect on basal OCR or non-mitochondrial respiration, although 10 ng/mL TNFα significantly reduced maximal OCR (Figure 5A). In contrast, cardiomyocytes from aged mice showed a decrease in maximal OCR with increasing TNFα concentration (Figure 5A). Furthermore, when comparing the response to 5 ng/mL TNFα, cardiomyocytes from aged mice exhibited significantly reduced basal OCR, maximal OCR and non-mitochondrial respiratory capacity, as well as maximal ECAR compared to those in cardiomyocytes from young mice (Figure 5B,C). These findings suggest that aging exacerbates TNFα-induced impairment of mitochondrial respiratory function in cardiomyocytes.

### 3.6. Impact of Aging on Mitochondrial Membrane Potential and OXPHOS in Mouse Cardiomyocytes Exposed to TNFα or LPS Stimulation

Mitochondrial membrane potential (ΔΨM), which maintains the proton gradient, is essential for mitochondrial respiration function and serves as an indicator of oxidative stress in cardiomyocytes. LPS and TNFα stimulation significantly reduced ΔΨM in cardiomyocytes from aged mice (Figure 6A), consistent with our previous observation in young cardiomyocytes [41]. We further assessed the protein levels of mitochondrial OXPHOS complexes. Aging significantly increased the expression of complex III subunit UQCRC2 in cardiomyocytes (Figure 6B). Additionally, complex I subunit NDUFB8 was significantly elevated in cardiomyocytes from aged mice compared to young mice after 1 h TNFα treatment (Figure 6B). Following LPS stimulation, aged cardiomyocytes exhibited significantly higher levels of complex I-NDUFB8, complex II-SDHB and complex V-ATP5A compared to those from young mice (Figure 6B). These results suggest that aging alters the mitochondrial OXPHOS response to TNFα or LPS stimulation in cardiomyocyte.

### 3.7. Changes in Signaling Molecules in Heart Tissue and Cardiomyocyte Following Inflammation in Aged Mice Compared to Young Adult Mice

Finally, we explored changes in additional proteins related to mitochondrial function in mouse heart tissue following CLP. Among these, the NADPH oxidase (NOX) family represents a main source of regulated ROS production. We observed that in heart tissue, aged mice—both in the sham-operated group and the CLP group, exhibited higher levels of NOX4 protein compared to their younger counterparts (Figure 7A1). In contrast, NOX2 protein expression did not differ between young and aged animals (Figure 7A2). Notably, CLP had no effect on myocardial expression of NOX2 or NOX4 in either age group. Stat3 signaling, which plays a key role in regulating inflammatory responses [42,43,44], was significantly activated by CLP, as indicated by increased levels of phosphorylated Stat3 (p-Stat3) in both young and aged mice hearts. This activation was more pronounced in the aged CLP group (Figure 7A3). These results suggest that aging differentially activates oxidative stress (via NOX4) and inflammatory signaling (via p-Stat3) in cardiac tissue during CLP-induced sepsis, with more sensitive responses observed in aged mice.

Given the heterogeneous cell population in the heart—including cardiomyocytes, cardiac endothelial cells, and cardiac fibroblast, we further examined alterations in these proteins specifically in cardiomyocytes following inflammatory stress. Similarly to the observations in whole-heart tissue, NOX4 protein levels were significantly elevated in aged cardiomyocytes compared to young cells, regardless of control or LPS treatment (Figure 7B1). Additionally, aged cardiomyocytes exhibited markedly increased NOX2 expression following LPS stimulation compared to young cardiomyocytes (Figure 7B2), while only a trend toward increased NOX2 was observed in the CLP group in aged mice (Figure 7A2). Similarly, neither TNFα nor LPS induced changes in NOX4 or NOX2 protein levels in cardiomyocytes from mice of either age group. In contrast to the patterns observed in whole-heart tissue, changes in p-Stat3 in cardiomyocytes differed notably (Figure 7B3). Aged cardiomyocytes showed higher baseline Stat3 activation compared to young cells. However, TNFα treatment did not increase p-Stat3 levels in either young or aged cardiomyocytes. Interestingly, while LPS stimulation significantly elevated p-Stat3 levels in young cardiomyocytes, no such increase was observed in aged cardiomyocytes (Figure 7B3).

## 4. Discussion

Myocardial dysfunction is one of major determinants for clinical outcomes in septic patients. Cardiac insufficiency characterized by reduced left ventricular contractility is seen in most patients with severe sepsis and septic shock [36,45,46]. Of note, the elderly population is more vulnerable to infections. The higher mortality rate of sepsis in older patients is associated with an increased incidence of septic cardiomyopathy compared to younger patients. Indeed, sepsis-induced myocardial injury is likely age-dependent [18]. While a trend towards decreased cardiac function was observed in aged mice compared to young animals in the sham groups, the difference was not statistically significant in the current research. This finding is not in line with the previous study showing both systolic and diastolic cardiac function declined in aged mice (24 months old) compared to young mice (3 months old) [47]. Such discrepancy may be due to the use of 19–21 months old mice in our work or may indicate that a larger sample size is needed to draw a more robust conclusion. Nevertheless, as expected, in this study, we observed that CLP significantly reduced cardiac function in both young and aged mice, with a higher incidence and greater severity of cardiac functional depression in the older group. Considering that the heart has high energy demands and cardiomyocytes are rich in mitochondria, this enhanced vulnerability may be due to dysfunctional mitochondria in the aged group.

We therefore explored this important unknown regarding whether impaired mitochondrial respiratory capacity existed in cardiomyocytes from aged mice. Indeed, mitochondrial dysfunction, specifically mitochondrial bioenergetic failure, has been recognized as important pathophysiological mechanisms of multiorgan dysfunction in sepsis patients [13]. In this study, we observed that cardiomyocytes from both young and old mice predominantly rely on mitochondria for aerobic respiration. In addition, our results clearly indicate the age-dependent changes in mitochondrial function in cardiomyocytes after exposure to LPS or TNFα. Mitochondrial respiratory capacity was decreased in cardiomyocytes derived from aged mice, which were more susceptible to inflammatory mediator-induced toxic effects compared to those from young adult mice. Accumulating evidence has indicated that mitochondrial abnormalities have been recognized as key factors in the pathogenesis of septic cardiomyopathy [48]. Dysfunctional mitochondria in senescent cells are likely unable to meet the increased metabolic demands imposed by sepsis [49], due to impaired respiratory chain function and decreased mitochondrial respiration efficiency in elderly individuals [50]. Our findings in cardiomyocytes from young and aged mice following inflammatory stress are consistent with these previous studies, supporting that reduced mitochondrial respiration likely contributes to the greater severity of cardiac functional depression in aged CLP mice. However, defective mitochondrial function in cardiomyocytes from aged mice does not impair ATP production under basal conditions, consistent with previous reports [51]. In terms of non-mitochondrial OCR, it reflects oxygen usage related to cellular metabolism occurring outside the mitochondria. We observed a significant reduction in non-mitochondrial OCR in cardiomyocytes from aged mice compared to young cells, accompanied by decreased glycolysis, as indicated by impaired maximal ECAR measurements. These findings suggest a potential decrease in overall metabolic activity or increased cellular stress, such as oxidative stress, in aged cardiomyocytes.

A normal mitochondrial membrane potential is essential for maintaining mitochondrial respiratory function [52]. Disruption of mitochondrial membrane potential leads to impaired mitochondrial bioenergetic profiles [53]. Our previous study showed that TNFα or H_2_O_2_ damages mitochondrial ΔΨm in cardiomyocytes from young mice, resulting in decreased maximal respiration [31,32]. In the current study, we observed a reduction in mitochondrial ΔΨm in cardiomyocytes from aged mice following exposure to either LPS or TNFα. This disruption likely collapses the proton gradient, thereby impairing the respiratory chain, OXPHOS activity, and overall cellular energy production. Consistent with this, we also observed a significant decrease in maximal OCR in these aged cardiomyocytes.

OXPHOS is the foundation of mitochondrial respiration, and alterations in any of the OXPHOS complexes can lead to impaired mitochondrial respiration. In the present study, we found that complexes I and IV were significantly decreased in the hearts of aged mice compared to younger mice without injury, whereas aging relatively unaffected myocardial expression of complexes II, III, and V, which is consistent with the results of other studies [54,55,56]. The reduced activities of complexes I and IV indicate a decreased capacity for ATP production via oxidative phosphorylation in the aged heart. Consistent with it, a decrease in mitochondrial respiration was noticed in cardiomyocytes from aged mice, implying a diminished ability to produce ATP (a reduced mitochondrial respiration efficiency). Importantly, following CLP, a significant increase in OXPHOS protein levels (all five complexes) was observed in the hearts of older mice, compared to both aged untreated controls and young CLP mice. Such elevated OXPHOS likely reflects a compensatory response to decreased mitochondrial respiration efficiency in the aged hearts, with increased ATP production needed to meet the demands of stress during sepsis. In addition, mitochondria from aged mouse hearts produced more ROS and exhibited increased oxidative damage compared to those from young hearts [57,58,59]. Therefore, this augmented OXPHOS may also suggest a greater potential for ROS generation and exacerbated oxidative damage [60]. Collectively, these alterations contribute to cardiac functional insufficiency after sepsis in aged mice.

In line with in vivo findings from aged mouse hearts, we observed a significant increase in OXPHOS protein levels in cardiomyocytes from aged mice treated with LPS (complexes I, II, III, and V) or TNFα (complexes I and III) compared to those from young mice. However, we did not observe an age-related reduction in OXPHOS protein levels (complexes I and IV) in untreated control cardiomyocytes, which contrasts with our findings in heart tissue and previous reports suggesting decreased stability of respiratory chain complexes with age [61]. This discrepancy may be attributed to the effects of in vitro cultivation on cardiomyocytes. Nevertheless, the differential expression of OXPHOS proteins between mouse heart tissue and cardiomyocytes warrants further investigation.

It is known that aging does not lead to changes in myocardial mitochondrial ultrastructure [62,63]. However, inflammation-induced alterations in mitochondrial cristae morphology and increased mitochondrial fission have been observed in several animal studies on sepsis [64,65,66]. Additionally, emerging evidence indicates reduced mitochondrial biogenesis, characterized by a lower number of mitochondria and decreased expression of biogenesis-related genes [67,68]. In the present study, we did not observe alterations in myocardial mitochondrial morphology in aged mice following CLP. Of note, two major types of cardiac mitochondria are identified: SSM—located under the plasma membrane and IFM—located between myogenic fibers [69]. It has been documented that IFM are less sensitive to ischemia/reperfusion (I/R) injury compared to SSM, likely due to their reduced exposure to oxygen gradients and greater tolerance to I/R-induced mitochondrial permeability transition relative to SSM [31,70]. However, it remains unclear whether IFM and SSM are affected differently in septic cardiomyopathy. In this study, we found CLP significantly reduced size of IFM in the hearts of aged mice, but not for SSM, suggesting a potentially decreased biogenesis in IFM. This finding is consistent with previous studies that CLP reduced mitochondrial size [71,72], and also reflects differences in the mitochondrial pathogenesis of septic cardiomyopathy compared to I/R injury.

Aging is associated with increased ROS production in the heart, particularly within mitochondria. NOX isoforms are major source of mitochondrial ROS generation. Among these, NOX2 (constitutively expressed, primarily located on the plasma membrane) and NOX4 (inducible, located on intracellular membranes, specifically in mitochondria), are the predominant isoforms in diseased myocardium and play key roles in mediating cardiac dysfunction [73,74]. Accumulating evidence indicates that mitochondrial impairment in septic cardiomyopathy leads to increased ROS levels [75,76]. Inhibition of NOX2 has been shown to reduce ROS production and improve cardiac function in animal models of sepsis [75]. However, in the present study, neither CLP nor aging changed myocardial NOX2 expression. In contrast, NOX4 expression was significantly increased in heart tissue and cardiomyocytes from older mice compared to young mice, in both treated and control groups. This finding aligns with previous studies showing that NOX4 expression in the heart increases with age [77]. Notably, NOX2 and NOX4 share only 39% homology [78]. The observed increase in NOX4 protein levels was not accompanied by changes in NOX2 protein levels [79], which may explain the differing age-related responses between NOX2 and NOX4 in the hearts and cardiomyocytes in this study. Interestingly, neither NOX2 nor NOX4 expression was affected by CLP or inflammatory mediators in the hearts and cardiomyocytes, raising questions about the role of NOX enzymes in sepsis-induced oxidative stress in the heart. Further investigation is needed to clarify this issue.

Extensive studies have shown a cytokine storm marked by increased levels of multiple inflammatory cytokines during the acute phase of sepsis. Therefore, we did not measure cytokine production in our study. Instead, we detected myocardial levels of Stat3, a key transcription factor in the JAK/STAT signaling pathway, which is involved not only in regulating inflammatory responses but also in modulating various cellular functions [80]. In fact, Stat3 plays a critical role in cardiomyocyte resistance to inflammation and acute injuries, and it is also implicated in the pathogenesis of age-related heart failure [81,82,83]. In this study, CLP significantly elevated myocardial Stat3 activation in aged mice, while an increased trend of Stat3 activation (p-Stat3) was observed in CLP-treated young mice. This finding is consistent with previous work indicating that the LPS-injected old mice showed higher levels of p-Stat3 compared to young animals [84]. It is well established that IL-6 plays a crucial role in mediating Stat3 activation, and age-related increases in plasma and cardiac IL-6 have been reported during endotoxemia [34,85]. Thus, elevated IL-6 levels may drive greater p-Stat3 in CLP-stressed aged mice compared to their younger counterparts. Given that Stat3 is constitutively expressed in cardiomyocytes, cardiac endothelial cells, fibroblasts, and smooth muscle cells [86,87], it is unsurprising to observe inconsistent patterns of Stat3 activation in cardiomyocytes in our study. With respect to the relationship between p-STAT3 and NOX activity, previous studies have shown that NOX4 is required for STAT3 activation in retinal capillary endothelia cells exposed to high glucose [88], and that the STAT3 pathway can regulate NOX4 expression in human vascular smooth muscle cells [89]. These findings suggest a positive correlation of Stat3 activation and NOX4 expression. Our results are consistent with these observations, as we found both increased NOX4 expression and enhanced Stat3 activation in aged hearts compared to young myocardium. However, future investigation is needed to elucidate the role of elevated Stat3 activation in myocardial responses, particularly in mitochondrial regulation and its interaction with NOXs, in the aged population during inflammation.

## 5. Conclusions

In conclusion, we confirmed a higher incidence and greater severity of cardiac functional insufficiency in aged mice subjected to CLP. Age-dependent mitochondrial impairments were observed in cardiomyocytes exposed to inflammatory stimuli, including reduced mitochondrial ΔΨm, disrupted mitochondrial respiratory function, and dysregulated OXPHOS complexes. Our findings highlight that the response of aging-impaired mitochondria to inflammation may underlie the worsened cardiac functional depression in the aged group during sepsis (Figure 8).

## Figures and Tables

**Figure 1 cells-14-01221-f001:**
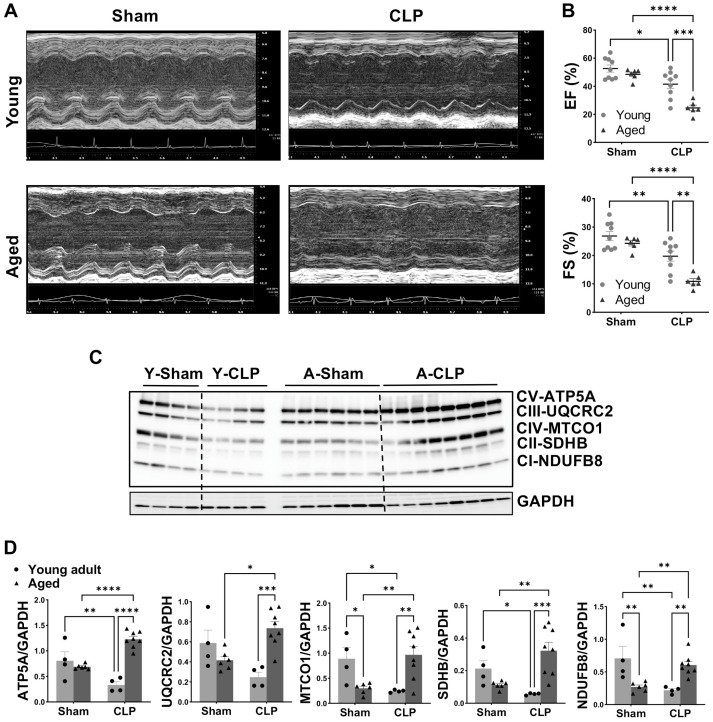
Worsened cardiac function and altered mitochondrial OXPHOS in aged mice compared to young animals following CLP. (**A**): Representative M-mode echocardiograph. (**B**): LV ejection fraction (EF) and LV fraction shortening (FS) in young adult and aged mice without or with CLP. (**C**): Myocardial protein levels of OXPHOS complexes [ATP5A-complex V (CV), UQCRC2-complex III (CIII), MTCO1-complex IV (CIV), SDHB-complex II (CII) and NDUFB8-complex I (CI)] detected by Western blot assay. (**D**): Quantification of the intensity of immunoblot bands—ATP5A, UQCRC2, MTCO1, SDHB, and NDUFB8 (normalized to GAPDH protein). Dots represent individual mouse heart value, mean ± SEM, two-way ANOVA, *n* = 4–8/group, * *p* < 0.05, ** *p* < 0.01, *** *p* < 0.0005, **** *p* < 0.0001.

**Figure 2 cells-14-01221-f002:**
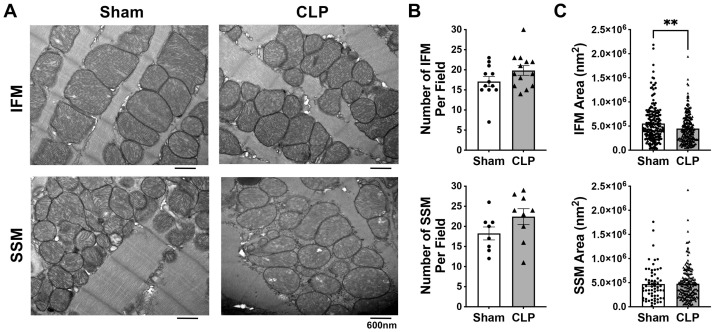
Myocardial mitochondrial ultrastructure in aged mice without or with CLP. (**A**): Representative electron micrographs of interfibrillar (IFM) and subsarcolemmal (SSM) mitochondria in the hearts of aged mice +/− CLP. Magnification: 23,000×. (**B**): Quantification of mitochondria number for IFM and SSM. (**C**): Quantification of mitochondria size for IFM and SSM. A total of ≥ 8 fields per condition were analyzed using the Image J software (NIH). Mean ± SEM, unpaired *t* test, ** *p* < 0.01.

**Figure 3 cells-14-01221-f003:**
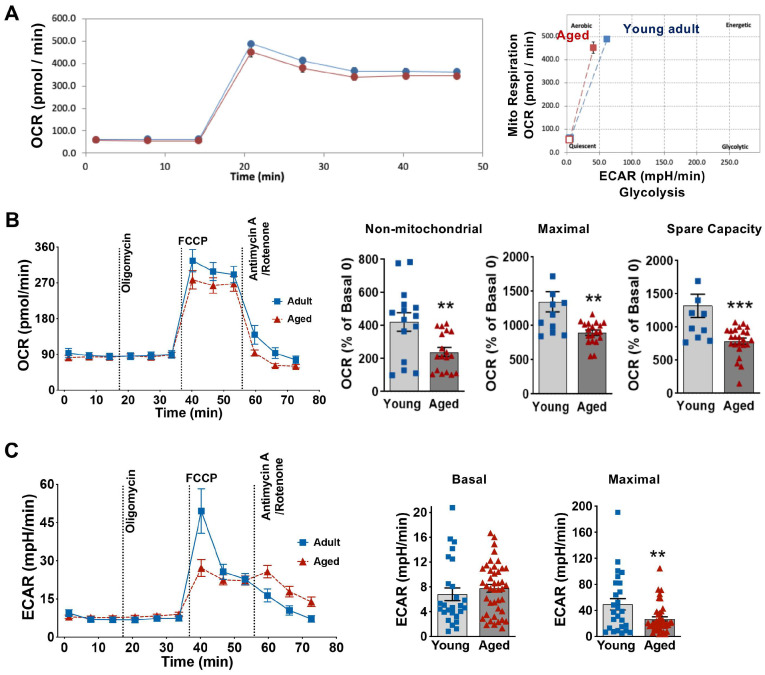
Aging on energy phenotype and mitochondrial metabolic function in cardiomyocytes. (**A**): Energetic phenotype of cardiomyocytes from young adult and aged male mice under basal and stress conditions (Blue: Young adult; red: aged). (**B**): The OCR trace following sequential injection of mitochondrial inhibitors (Oligomycin, FCCP, Antimycin A, and Rotenone) and parameters of the non-mitochondrial respiration, maximal respiration, and spare respiratory capacity in adult and aged cardiomyocytes. (**C**): Changes in ECAR in adult and aged cardiomyocytes exposed sequentially to different modulators of mitochondrial activity. Data are from *n* = 3–5 mouse hearts/group, mean ± SEM, unpaired *t* test, ** *p* < 0.01 and *** *p* < 0.0005. OCR: oxygen consumption rate; ECAR: extracellular acidification rate.

**Figure 4 cells-14-01221-f004:**
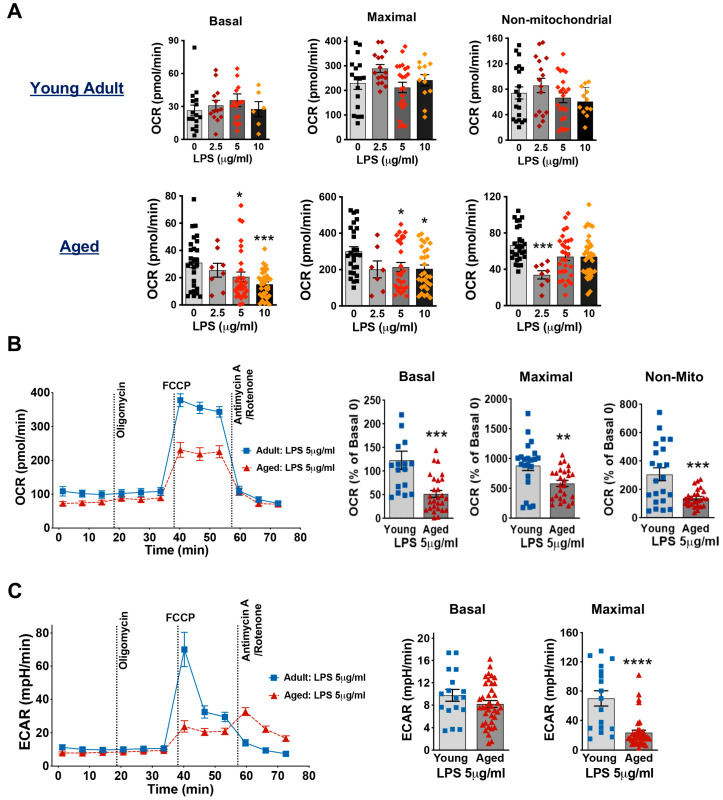
Mitochondrial respiratory function in cardiomyocytes from young adult and aged mice when exposed to LPS. (**A**): Mitochondrial OCR in young adult (11–21 weeks) and aged (20–21 months) mouse cardiomyocytes treated with LPS at 0, 2.5, 5, and 10 μg/mL for 1 h. Shown are basal, maximal, and non-mitochondrial respiration. (**B**): The OCR trace in cardiomyocytes from young adult and aged mice treated with 1 h LPS at 5 μg/mL (**left**) and the parameters of basal, maximal and non-mitochondrial (non-Mito) respiration (**right**), represented as a percentage of the Basal OCR in the group without LPS (0 μg/mL LPS). (**C**): The ECAR data in young adult and aged cardiomyocytes following exposure to 5 μg/mL LPS for 1 h. Mean ± SEM, *n* = 4–5 hearts/group, * *p* < 0.05, ** *p* < 0.01, *** *p* < 0.0005, and **** *p* < 0.0001.

**Figure 5 cells-14-01221-f005:**
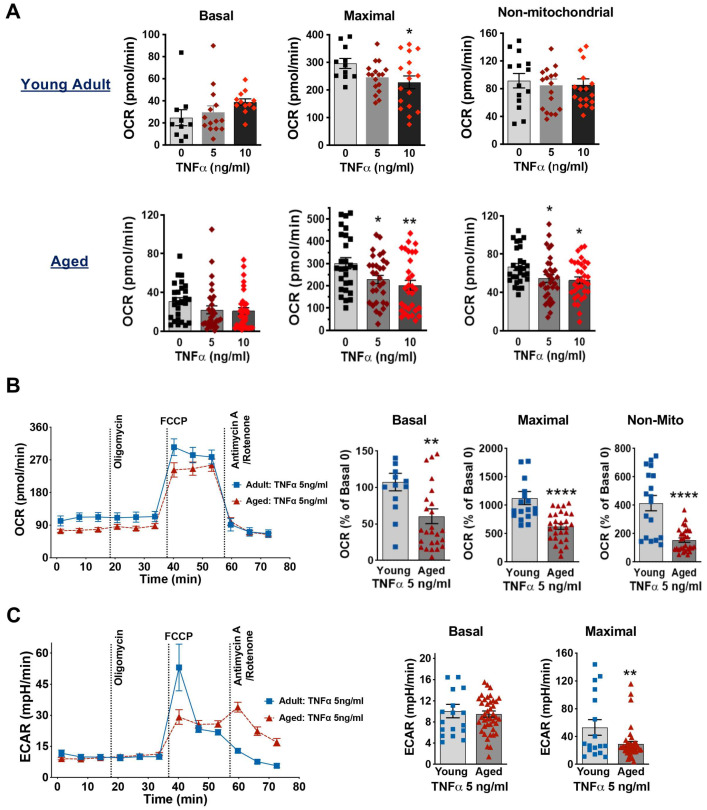
Mitochondrial bioenergetic profiles in cardiomyocytes from young adult and aged mice when exposed to TNFα. (**A**): Mitochondrial OCR in young adult (11–21 weeks) and aged (20–21 months) mouse cardiomyocytes treated with different concentrations of TNFα (0, 5, and 10 ng/mL) for 1 h. Measurements of basal, maximal, and non-mitochondrial respiration are shown. (**B**): The OCR trace of cardiomyocytes from young adult and aged mice treated with 1 h TNFα (5 ng/mL) (left) and their basal, maximal and non-mitochondrial (non-Mito) respiration (right), expressed as % of the Basal OCR in the group without TNFα (0 ng/mL TNFα). (**C**): The ECAR in young adult and aged cardiomyocytes following exposure to 5 ng/mL TNFα for 1 h. Mean ± SEM, *n* = 4–5 hearts/group, * *p* < 0.05, ** *p* < 0.01, **** *p* < 0.0001.

**Figure 6 cells-14-01221-f006:**
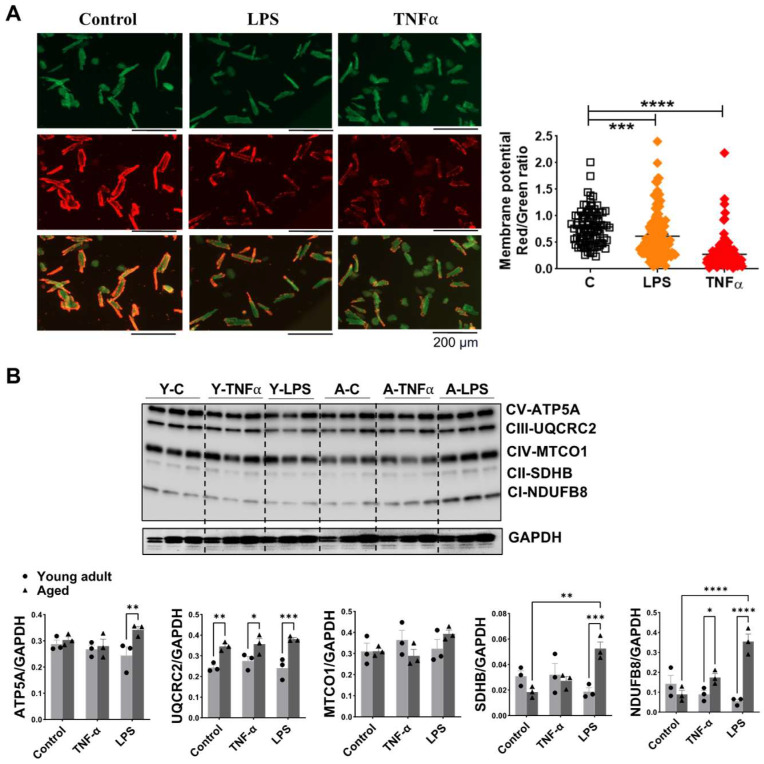
Aging on mitochondrial membrane potential and OXPHOS in mouse cardiomyocytes exposed to TNFα or LPS stimulation. (**A**): Representative images of JC-1-stained cardiomyocytes from aged mice, treated with 1 h LPS or TNFα. Mitochondrial membrane potential was quantified by the ratio of fluorescence intensity of red to green in individual cardiomyocytes. (**B**): Protein levels of OXPHOS complexes [ATP5A-CV, UQCRC2-CIII, MTCO1-CIV, SDHB-CII and NDUFB8-CI] in cardiomyocytes from young adult and aged male mice after exposure to 1 h TNFα and LPS, using GAPDH as an internal reference. Mean ± SEM, *n* = 3 individual experiments/group, two-way ANOVA. * *p* < 0.05, ** *p* < 0.01, *** *p* < 0.0005, **** *p* < 0.0001.

**Figure 7 cells-14-01221-f007:**
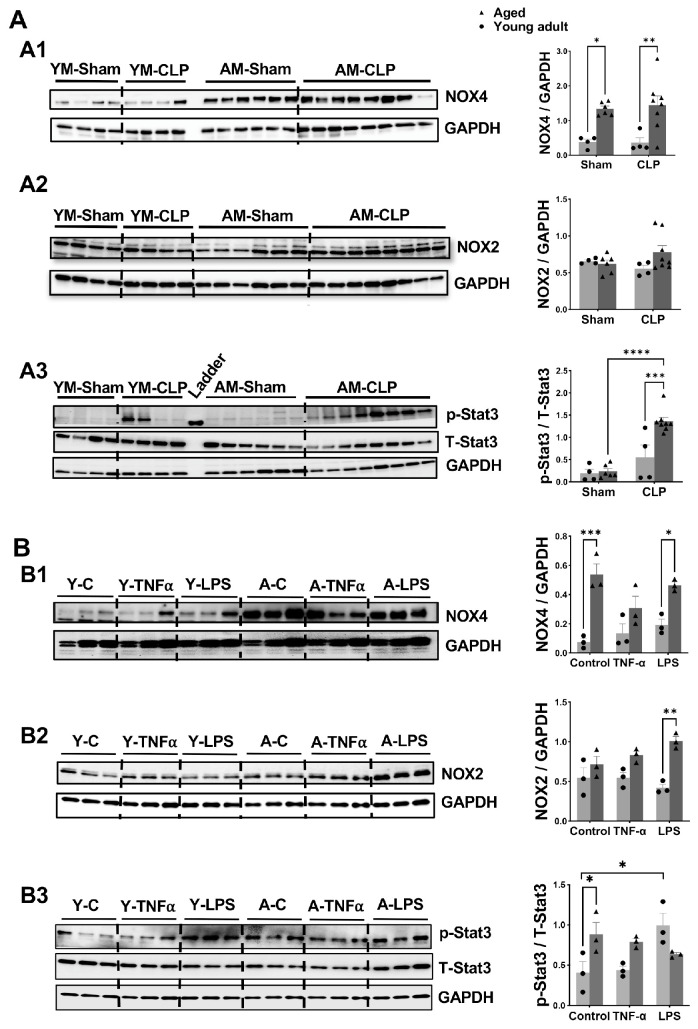
Changes in signaling molecules in heart tissue and cardiac myocyte following inflammation in aged mice compared to young adult mice. (**A**): Western blot assay of NOX4 (A1), NOX2 (A2), phosphorylated (p-) and total (T-) of Stat3 (A3) protein levels in heart tissue from young adult and aged male mice following CLP. (**B**): Protein expression of NOX4 (B1), NOX2 (B2), p-Stat3 and T-Stat3 (B3) in the cardiomyocytes from young adult and aged male mice after exposure to TNFα and LPS, normalized to GAPDH. Mean ± SEM, *n* = 3–8 hearts/group in A, *n* = 3 individual experiments/group in B, two-way ANOVA. * *p* < 0.05, ** *p* < 0.01, *** *p* < 0.0005, **** *p* < 0.0001.

**Figure 8 cells-14-01221-f008:**
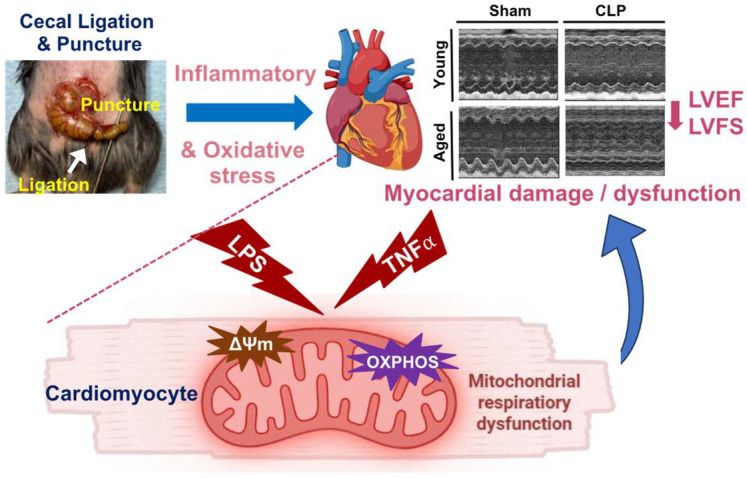
A proposed mechanism suggests that aging-related impairment of mitochondrial membrane potential (ΔΨm), disrupted mitochondrial respiratory function, and dysregulated OXPHOS complexes contribute to worsened cardiac insufficiency in aged mice during sepsis. Part of this figure was created by the author using Biorender (biorender.com).

## Data Availability

Data is contained within the article.

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
