# Peer review of "Age-Related Mitochondrial Alterations Contribute to Myocardial Responses During Sepsis"

_cells, 2025, doi:10.3390/cells14151221_

Round 1
Reviewer 1 Report
Comments and Suggestions for Authors
Aging is a significant risk factor for sepsis-related morbidity and mortality; however, the underlying mechanisms remain incompletely understood. In this study, Du et al. investigated how aging contributes to mitochondrial metabolic impairment and cardiac dysfunction during sepsis. They demonstrated that aging exacerbates cardiac dysfunction and alters mitochondrial oxidative phosphorylation (OXPHOS) in septic conditions. Mechanistic studies revealed that aged mice exhibited decreased mitochondrial respiratory function and increased susceptibility to inflammatory toxicity compared to young adult mice. Notably, a significant increase in OXPHOS protein levels was observed in the hearts of aged mice, suggesting a potential compensatory response to impaired mitochondrial metabolism and a heightened capacity for reactive oxygen species (ROS) generation during sepsis.
This study presents compelling findings regarding the role of aging in mitochondrial dysfunction and cardiac impairment during sepsis. The manuscript is well written, presenting both positive and negative results with clear explanations. The conclusions are well supported by robust experimental evidence, and the results are clearly and logically presented. This is a solid and meaningful contribution to the field.
There are no major concerns that need to be addressed. However, minor typographical errors should be corrected—for example, in line 270, the statistical notation "***p < 0.005" should be revised to align with the standard conventions used elsewhere in the manuscript.
Reviewer 2 Report
Comments and Suggestions for Authors
Authors Du et al investigated the impact of aging on trauma and cellular mechanism using a translational animal model and integrative cellular approaches. The group presents very interesting results in which aging significantly contributes to the outcome of animal exposed to CLP challenging, which is highly related to different mitochondrial complexes (oxidative phosphorylation) and mitochondrial respiratory rates. In addition, authors proceeded to examine whether there existed the differences of inflammatory stress induced by either LPS or TNF, they noticed that cardiomyocytes from young and aged animals manifested the different response as indicated by reading out including OCR via measurement of seahorse extra cellular flux in vitro studies. Authors also showed that NOXs and STAT3 act as one of major pathways that are attributable to ageing-induced the difference following an exposure to stresses. The study was well designed, and results are solidly presented to reveal novel pathways using aged animals and preclinical studies. This will advance the knowledge in the field of traumatic disease and provide innovative information. Nevertheless, there are a few points that could be addressed more to help understand more clearly.
- Please discuss whether there is a difference in cardiac function at the basic condition such as sham conditions between the young and aged group.
- The further explanation of no-mitochondrial OCR could be briefly expanded to help the understanding of the difference of OCR between mitochondrial and no-mitochondrial components in the section of discussion or method.
- Please also discuss or cite any correlation of STAT3 phosphorylation with NOX2 or NOX4 if possible, to support the results.
Reviewer 3 Report
Comments and Suggestions for Authors
Several major concerns needs to be addressed .
- Please provide the distance and time scale in Figure 1A.
- LVEF and LVFS are not sufficient to display the cardiac systolic function, please clarify.
- Survival ratio is an important index to show the severity of sepsis, please examine the survival ratio in Figure 1.
- Interesting,the author showed no difference of the number and mass of mitochondria in aged mice with or without CLP. Previous studies have shown CLP can induced mitochondrial damage in heart tissue in mice, please discuss.
- Please also provide the ECAR data in Figure 3 and Figure 4.
Round 2
Reviewer 3 Report
Comments and Suggestions for Authors
The authors have well addressed my concerns, I have no further comments.